# QUANTIFYING CLASSIFICATION PERFORMANCE THROUGH COMBINATORIAL GEOMETRY AND LOCALIZED DATA ANALYSIS

## ABSTRACT

Understanding the theoretical boundaries of a learning mechanism and ascertaining its fundamental capabilities remains a persistent challenge in machine learning. While the VC-dimension has been instrumental in quantifying a model's data-fitting abilities, its independence from data distribution sometimes limits its practicality. In this study, we address the problem of establishing realistic bounds on a model's classification power by harnessing the underlying combinatorial geometry of data using novel tools. We introduce conditions that rely on *local* computations performed on small data subsets to determine the *global* performance of classifiers. Specifically, by considering a dataset $\{(X_i, y_i)\}_{i=1}^n$, where $X_i \in \mathbb{R}^d$ is a feature vector and $y_i$ is the corresponding label, we establish optimal bounds on the training error (in terms of number of misclassifications) of a linear classifier based on the linear separability of local data subsets, each comprising of $(d + 2)$ data points. We also prove an optimal bound on the margin of Support Vector Machines (SVMs) in terms of performance of SVMs on $(d + 2)$ sized subsets of data. Furthermore, we extend these results to a non-linear classifier employing hypersphere boundary separation. This research contributes valuable insights into assessing the classification potential of both linear and non-linear models for large datasets. By emphasizing local computations on subsets of data with fixed cardinality, it provides a foundation for informed and efficient decision-making in practical machine learning applications.

## 1 INTRODUCTION

*How can we rigorously evaluate the fundamental capabilities of a learning mechanism?* This question remains an enduring challenge in machine learning theory, particularly given the field's transformative impact on various domains in recent years. One of the most significant contributions to this issue is the concept of VC-dimension (Vapnik–Chervonenkis dimension) Vapnik (2000); Harvey et al. (2017); Blumer et al. (1989). This measure provides theoretical bounds on a classifier's performance by representing the maximum number of data points a model can 'shatter', given all possible arrangements and labelings. Essentially, VC-dimension gauges a model's ability to fit even the most complex patterns within a dataset. For instance, consider a binary classification problem where we seek to draw an optimal decision boundary to separate data points into two (positive and negative) classes. The VC-dimension of a hypothesis class or model reveals the largest number of data points this model can correctly classify, irrespective of their arrangement, thus encapsulating the essence of a model's learning prowess.

Despite its versatility, the VC-dimension does not account for the distribution of data points, which sometimes limits its use as a practical measure to quantify the classification power of classifiers Steinke & Zakynthinou (2020); Holden & Niranjan (1995); Kowalczyk & Ferrá (1996); Lorena et al. (2019); Mossel & Umans (2002). For example, the VC-dimension of a linear classifier is $d + 1$, i.e., there exists a set of $d + 2$ points with a particular labeling for which a linear classifier can not learn the decision boundary of a binary classification problem. However, a linear classifier effectively and successfully classifies large dataset of points. The fact that some rare prohibitive point configurations exist is of little consequence in practice. Therefore, it is more desirable from a practical viewpoint to design a satisfactory bound that takes into account the arrangement of data.

To elaborate on this, consider a scenario where $\mu$ is an unknown probability distribution over a product set $\mathcal{X} \times Y$. Here, $\mathcal{X}$ is a metric space of (potentially very large) dimension $d$, representing the feature vectors of data points (henceforth, referred to as data points) and $Y = \{-1, +1\}$ denotes the class labels. Let $(X_1, y_1), (X_2, y_2), \ldots, (X_n, y_n)$ be independent samples drawn from $\mu$. For simplicity, we focus on a binary classification problem, where $y_i \in \{\pm 1\}$. Within this framework, we address the following question: **can one measure a realistic bound on the classification power of a classifier on this sample "efficiently"**. We assume that $n$ is large enough to prohibit the *global* processing of the complete dataset at once. Instead, the processing relies on numerous *local* queries consisting of a small fraction of data points. Further, the sampling process might be noisy, making it impractical to train a model on these samples alone. Also, we focus mostly on linear classifiers in this paper. With these settings in mind, we ask the following question: For a given set of samples $(X_i, y_i)$ from an unknown distribution $\mu$, is it possible to deduce from local computations on small subsets of samples whether there exists a linear classifier that correctly classifies every data point? The answer to this question is in the affirmative. Informally, we know that if the data points are linearly classifiable *locally*, then they are also classifiable *globally*. This conclusion stems from Kirchberger's famous theorem in discrete geometry, proven in 1903 (Kirchberger (1903); Webster (1983); Houle (1991); Lay (2007)).

**Theorem 1.** *(Kirchberger 1903) Given that $\mathcal{A}$ and $\mathcal{B}$ are compact subsets of Euclidean space, $E^d$, then for every subset $T \subseteq \mathcal{A} \cup \mathcal{B}$, with $|T| \leq d + 2$, $T \cap \mathcal{A}$ and $T \cap \mathcal{B}$ can be strictly separated by a hyperplane if and only if $\mathcal{A}$ and $\mathcal{B}$ can be strictly separated by a hyperplane.*

Here, local computation refers to the task of evaluating whether a given $(d + 2)$-sized subset of the dataset is linearly separable. Importantly, this computation size independent of the overall dataset size $n$. For instance, if one has a dataset comprising of a large number of data points in a reasonably large dimension $d$, the theorem indicates that the question of linear separability for the entire dataset can be decided by examining multiple subsets of just $d + 2$ points. Moreover, these computations can be naturally parallelized, offering efficiency in practical applications.

In this work, we extend the above mentioned result in multiple significant directions. First, we consider the scenario where data is linearly separable. In such cases, the Support Vector Machine (SVM) Cervantes et al. (2020); Campbell & Ying (2022); Tan & Wang (2004); James et al. (2013) serves as a canonical linear classifier. Specifically, SVM aims to find the classifier that maximizes the distance—known as the 'margin'—between the separating hyperplane and the nearest data points (the closest/support vectors). The associated optimization problem is:

$$\text{Minimize:} \quad \frac{1}{2}\|\mathbf{w}\|^2$$
$$\text{Subject to:} \quad y_i(\mathbf{w} \cdot X_i + b) \geq 1, \quad \text{for } i = 1, 2, \ldots, n$$

where, $\mathbf{w}$ represents the weight vector, $b$ represents the bias term, and $X_i$ and $y_i$ are as previously defined. The *margin*, is calculated as $\frac{1}{\|\mathbf{w}\|}$. We address the following question regarding the value of margin an SVM algorithm may achieve for a particular dataset.

Given a linearly separable set of samples $(X_i, y_i)$ and a constant $w_0$, is there a hyperplane $\mathbf{w}^T X - b = 0$ such that, $y_i(\mathbf{w} \cdot X_i + b) \geq 1$, and the $\frac{1}{\|\mathbf{w}\|} \geq w_0$.

In the context of this question, and in the spirit of the Kirchberger theorem, we prove:

**Theorem 2.** *(SVM-Kirchberger) Let $\mathcal{A}$ and $\mathcal{B}$ be disjoint, non-empty compact sets of $E^d$. Then $\mathcal{A} \cup \mathcal{B}$ is strictly linearly separable with margin $w_0$ if and only if for each subset $T \subset \mathcal{A} \cup \mathcal{B}$ of $d+2$ or fewer points, there exists a linear SVM of margin $w_0$ that strictly separates $T \cap \mathcal{A}$ and $T \cap \mathcal{B}$.*

To address a more real-world scenario where the data may not be perfectly linearly separable, we introduce a significant advancement: a *fractional extension of the Kirchberger theorem*. This fractional version offers a way to quantify the margin of error—specifically, it provides a bound on the number of misclassified samples based on local data of $(d+2)$ samples. As a result, this allows us to infer global property (misclassification) of the dataset based on local $(d + 2)$-sample data sets. The formal statement of our result is as follows:

**Theorem 3.** *(Fractional Kirchberger) Let $\alpha$ be a constant in the range $0 < \alpha \leq 1$. Consider a dataset $\mathcal{A} \cup \mathcal{B} \subset E^d$, with $|\mathcal{A} \cup \mathcal{B}| = n$. If an $\alpha$ fraction of all $(d+2)$-member subsets of $\mathcal{A} \cup \mathcal{B}$ are*

*strictly linearly separable, then there exists a constant $\beta$, such that at least $\beta n$ members of $\mathcal{A} \cup \mathcal{B}$ are also linearly separable. Moreover, $\beta \geq 1 - (1 - \alpha)^{1/(d+1)}$, and this bound on $\beta$ is optimal.*

Proceeding further, we extend the fractional Kirchberger theorem into the realm of SVMs, and show the following result.

**Theorem 4.** *(SVM Fractional Kirchberger) Let $\mathcal{A}$ and $\mathcal{B}$ be disjoint, non-empty compact sets of $E^d$, with $|\mathcal{A} \cup \mathcal{B}| = n$. Given a dimension $d$ and $\alpha > 0$, assume that an $\alpha$ fraction of the $(d+2)$-member subsets of $\mathcal{A} \cup \mathcal{B}$ can be linearly classified by an SVM algorithm with a margin of $w_0$. Then, there exists a constant $\beta(\alpha, d) > 0$ such that $\beta n$ members of $\mathcal{A} \cup \mathcal{B}$ can be accurately classified using a soft-margin SVM with a margin of $w_0$.*

To clarify, a soft-margin SVM modifies the original constraints, allowing for some degree of misclassification. Specifically, the constraints become, $y_i(w \cdot X_i + b) \geq 1 - \xi_i$, and the objective function includes an additional penalty term, $C \sum_{i=1}^{n} \xi_i$, to account for each misclassified point. Finally, we apply our methods to the case of *separation by hypersphere* in $d$-dimensions as an example of a non-linear classifier. We prove a fractional Kirhberger type theorem for the hypersphere separation.

**Theorem 5.** *(Fractional Hypersphere Separation) For a constant $\alpha < 1$, and finite disjoint point sets $\mathcal{A}, \mathcal{B} \subset E^d$, if $\alpha \binom{n}{d+3}$ of the distinct $d + 3$-member subsets of $\mathcal{A} \cup \mathcal{B}$ are strictly spherically separable, then there exists a constant $\beta < 1$ such that $\beta n$ points of $\mathcal{A} \cup \mathcal{B}$ are strictly separated by a hypersphere.*

In addition to the theoretical contributions highlighted earlier, we also perform an experimental evaluation. Our methods are versatile, capable of handling real-world complications like imprecise sensors and data corruption, making them both theoretically and practically relevant.

The rest of the paper is organized as follows: Section 2 introduces preliminaries and reviews the idea of point-hyperplane duality that is used throughout the paper. Section 3 is the main section and presents proofs of Theorems 2, 3, and 4, which state Kirchberger type results for SVMs. Section 2 discusses the case of hypersphere separation and proves Theorem 5. Finally, Section 5 concludes the paper.

## 2    PRELIMINARIES AND BACKGROUND

In this section, we present preliminaries for understanding the key concepts that underpin our results. We begin by introducing point-hyperplane duality, a fundamental concept in our proofs, which elegantly transforms points into flat affine subspaces and vice versa. This concept is widely known in the literature as point-hyperplane duality Agarwal & Sharir (2005); Bennett & Bredensteiner (2000). To provide a clear understanding, we include a concise introduction with a small illustrative example in Section 2.1. Our results also draw upon the fractional versions of the well-known Helly theorem. To contextualize these theorems, we briefly review the the Fractional Helly Theorem in Section 2.2. For further details, we refer the readers to Matousek (2002). Before discussing these prerequisites, we define some preliminary notions that will be employed throughout this paper. As stated in the introduction, the subject of this paper will be data points that are represented as $d$-tuples of real numbers in Euclidean space $E^d$. Thus, feature vectors are points in $E^d$.

**Definition 1.** *(Hyperplane)- A hyperplane $h$ in $E^d$ is defined for real coefficients $a_1, a_2, .., a_d, a_{d+1}$, not all identically equal to 0, as:*

$$h = \{x \in E^d : a_1 x_1 + a_2 x_2 + .. + a_d x_d + a_{d+1} = 0\}.$$

**Definition 2.** *(Signed Halfspace)- For hyperplane $h \in E^d$, the positive open half-space $h^+$ is defined as:*

$$h^+ = \{x \in E^d : a_1 x_1 + a_2 x_2 + .. + a_d x_d + a_{d+1} > 0\},$$

*and the negative open half-space $h^-$ is defined as:*

$$h^- = \{x \in E^d : a_1 x_1 + a_2 x_2 + .. + a_d x_d + a_{d+1} < 0 \}.$$

**Definition 3.** *(Strict Linear Separability)- Let $\mathcal{A}$ and $\mathcal{B}$ be disjoint point sets in $E^d$. If $\mathcal{A}$ and $\mathcal{B}$ are strictly linearly separable, then then there exists a hyperplane $h$ and associated open half-spaces $h^+$ and $h^-$, such that $\mathcal{A} \subset h^+$, $\mathcal{B} \subset h^-$.*

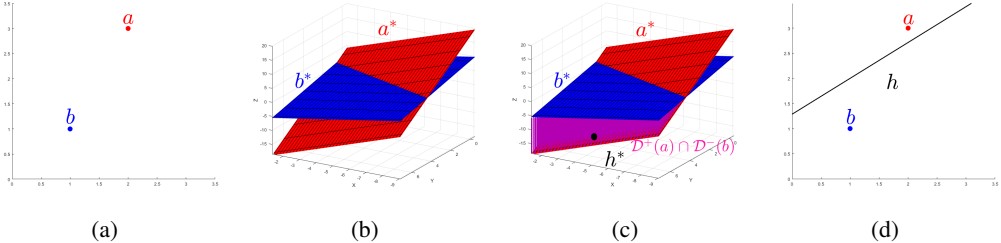

Figure 1: (a) Points $a$ and $b$ in primal space $E^2$. (b) $a^*$ and $b^*$ in dual space $E^3$. (c) $h^* = (-5, 7, -9)$ is a point in the intersection of $\mathcal{D}^+(a)$ and $\mathcal{D}^-(b)$. (d) $h$ is a linear separator for points $a$ and $b$.

The term *fractional separability* will be used in this paper to refer to the ratio of correctly classified data points to the cardinality of the dataset achieved by an optimal classifier in the sense that the this ratio is maximal over all possible classifiers of a given type.

## 2.1 POINT-HYPERPLANE DUALITY

In the following sections, we will explore a powerful concept known as duality transformation between points and hyperplanes. When dealing with a set of points in a Euclidean space, referred to as the *primal space*, we can create another Euclidean space, the *dual space* or reference space. In this dual space, a unique relationship exists between points in the primal space and hyperplanes in the dual space, and vice versa. This duality transformation has two essential qualities: (a) *Preservation of incidences:* It ensures that the relationships or incidences between points and hyperplanes remain intact. (b) *Consistency in order:* It maintains the order of incidences, which can either be identical or opposite to that in the primal or reference space. This latter quality is particularly important for connecting separating hyperplanes to points in set intersections. Much of the literature on geometric duality focuses on the point-line duality mapping, denoted as $\pi : E^2 \mapsto E^{2*}$. In this mapping, each point $p$ (or line $l$) in the primal space corresponds to a line $p*$ (or point $p*$) in the dual space, respectively, with the same incidence and order properties as mentioned earlier. In this paper, we adopt similar notation and language to describe duality transformations in $E^d$. To facilitate understanding, we introduce preliminary definitions and notation related to point-hyperplane duality, followed by an illustrative example.

**Definition 4.** *(Duality Transform) We define the duality transform $\mathcal{D} : E^d \mapsto E^{d+1}$ in the following manner for point $p = (p_1, p_2, .., p_d) \in E^d$, and hyperplane $h = \{x \in E^d : a_1 x_1 + a_2 x_2 + .. + a_d x_d + a_{d+1} = 0\}$:*

$$\mathcal{D}(p) = p^* = \{x \in E^{d+1} : x_1 p_1 + x_2 p_2 + .. + x_d p_d + x_{d+1} = 0\},$$
$$\mathcal{D}(h) = h^* = (a_1, a_2, .., a_{d+1}) \in E^{d+1}.$$

*Example:* Consider points $a = (2, 3)$ and $b = (1, 1)$ in $E^2$ (Figure 1). We can find the equation of a separating hyperplane in the following way:

(i) The signed duality transform of $a$ yields:
$$\mathcal{D}^+(a) = a^{*+} = \{x \in E^3 : 2x_1 + 3x_2 + x_3 > 0\}.$$

(ii) The signed duality transform of $b$ yields:
$$\mathcal{D}^-(b) = b^{*-} = \{x \in E^3 : x_1 + x_2 + x_3 < 0\}.$$

(iii) Select a point $h^*$ such that $h^* \in \mathcal{D}^+(a) \cap \mathcal{D}^-(b)$. The point $h^* = (-5, 7, -9)$ satisfies this condition.

(iv) Take the dual transformation of $h^*$ to produce $\mathcal{D}(h^*) = h = \{x \in E^2 : -5x_1 + 7x_2 - 9 = 0\}$.

(v) $h$ strictly linearly separates $a$ and $b$.

**Set Notation** – Instead of adhering to the traditional classification language used in the introduction, we opt for the convenience of separate labels, denoted as $\mathcal{A}$ and $\mathcal{B}$, to represent sets of data points belonging to the positive class ($y_i = +1$) and the negative class ($y_i = -1$), respectively. Subsequently, the upcoming theorems will employ the notation of $\mathcal{A}$ and $\mathcal{B}$ without any loss of generality.

## 2.2 Fractional Helly

'Helly-type' problems broadly address the global properties of a set of geometric objects that are implied by properties of its local subsets, typically of some fixed cardinality Helly (1923); Eckhoff (1993); Bárány et al. (1982); Bárány & Kalai (2022). The crux of the arguments presented in Section 3 will be based on a fractional variety of Helly's Theorem. Our exploration centers on scenarios where not all subsets of cardinality $(d + 1)$ or fewer share a common intersection. Instead, only a fraction, denoted as $\alpha$, exhibits this property. Throughout this paper, we will refer to the fraction of the entire set with a common intersection as $\beta$. This concept was first elucidated by Liu and Katchalski Katchalski & Liu (1979), who demonstrated that for a given value of $\alpha$, a lower bound can be guaranteed for $\beta$. Subsequently, Kalai proved an optimal value of $\beta$ in terms of $\alpha$ and $d$ as an application of his work on the Least Upper Bound Theorem G.Kalai (1984), which we state here and refer to as the Fractional Helly Theorem:

**Theorem 6.** *Let $\mathcal{F}$ be a finite family of convex sets in $E^d$ with $|\mathcal{F}| = n$. If at least $\alpha$ of the $\binom{n}{d+1}$ subsets of size $d + 1$ have non-empty intersection, where $\alpha \in [0, 1]$, then at least $\beta(\alpha, d) > 0$ members of $\mathcal{F}$ have a common intersection. In particular, $\beta = 1 - (1 - \alpha)^{1/(d+1)}$.*

The derivation of this bound hinges on a fascinating insight involving the representation of a family of convex sets and their intersection patterns as a simplicial complex, often referred to as a 'nerve', within $E^d$. In this context, each set is represented as a vertex, and intersections between sets are depicted as edges. Notably, this bound is demonstrated to be optimal in the general case. However, the lower bound in Theorem 6 provides an asymptotic estimate $n \to \infty$. For a specific value of $n$, a more precise version of the Fractional Helly Theorem is implied by a result known as the 'Least Upper Bound Theorem' due to Kalai G.Kalai (1984) and Eckhoff Eckhoff (1985).

**Theorem 7.** *Let $\mathcal{F}$ be a family of convex sets in $\mathbb{E}^d$, with $|\mathcal{F}| = n$. If there are $\alpha\binom{n}{k}$ intersecting $k$-tuples of $\mathcal{F}$, and $\alpha\binom{n}{k} > \sum_{i=0}^{d} \binom{r}{k-i}\binom{n-r}{i}$, then $\mathcal{F}$ has an intersecting subfamily of size at least $d + r + 1$.*

This result yields a tight bound on $\beta$ when the size of $\mathcal{F}$ is given in addition to the value of $\alpha$. In the real world scenarios we encounter, this value of $\beta$ is most valuable.

## 3 Kirchberger-Type Theorems for SVMs

In this section we present the proofs of the theorems stated in the introduction. We begin with the proof of Theorem 2, an extension of Kirchberger's Theorem to the realm of strict linear separation with a margin in Section 3.1. Subsequently, we present the proof of the fractional version of Kirchberger's theorem, Theorem 3, in Section 3.2. Recall that 'fractional' signifies scenarios where only a fraction of subsets of size $d+2$ samples are linearly separable. Following the proof, we present some simulation results for illustration. Moving forward in Section 3.3, we give the proof of Theorem 4, which provides a lower bound on the performance of a soft-margin SVM classifier—a fractional counterpart to Theorem 2. A notable feature unifying all the theorems presented in this section, and one that the forthcoming proofs substantiate, is their remarkable dependency solely on fundamental dataset properties: size, dimensionality, and the average performance of the classifier on subsets of cardinality $(d + 2)$.

### 3.1 Hard-Margin SVM Classification

Here we present a proof that establishes a crucial equivalence: the local conditions required for ensuring perfect linear separation with a specified margin, specifically the distance between a separating hyperplane and the closest data points to that hyperplane, remain identical to those articulated in Kirchberger's theorem. The key distinction lies in replacing 'linearly separable' with 'linearly separable with a margin of $w_0$(Figure 2). This leads us to a significant conclusion: Kirchberger's theorem applies not only to strict linear separation but also to strict linear separation with a margin, making it directly applicable to support vector machines. We note that this result is mentioned in Lay (2007) without a proof. This result will also play a pivotal role in proving Theorem 4.

*Proof.* **Theorem 2 (SVM-Kirchberger)**: It is enough to prove the sufficiency of the condition. Let $U$ denote a $(d+2)$-member subset of $\mathcal{A} \cup \mathcal{B}$. Since each $(d+2)$-point subset $U$ of $\mathcal{A} \cup \mathcal{B}$ is linearly classifiable with margin $w_0$ then for all such $U$ there is a hyperplane $\mathbf{w}^T X - b = 0$, such that ($\mathbf{w} \cdot a \leq 1 | a \in U \cap \mathcal{A}$) and ($\mathbf{w} \cdot b \geq 1 | b \in U \cap \mathcal{B}$), with $\frac{1}{\|w\|} \geq w_0$. These conditions are equivalent to the following: $\forall u \in U, \forall x \in \mathbf{w}, \min(d(u,x)) \geq \frac{w_0}{2}$ and $\forall a \in U \cap \mathcal{A}, a \in \mathbf{w}^-, \forall b \in U \cap \mathcal{B}, b \in \mathbf{w}^+$, where $\mathbf{w}^{+/-}$ is the positive/negative halfspace bounded by $\mathbf{w}$. Consider the modified sets $\mathcal{A}'$ and $\mathcal{B}'$ such that $\mathcal{A}' = \mathcal{A} + \{x \in E^d : \|x\| < \frac{w_0}{2}\}$ and $\mathcal{B}' = \mathcal{B} + \{x \in E^d : \|x\| < \frac{w_0}{2}\}$. In other words, the modified sets consist of open balls of radius $\frac{w_0}{2}$ centered at the points of the original set. For each $U'$, where $U'$ is a $(d+2)$-member subset of $\mathcal{A}' \cup \mathcal{B}'$, $(\min(d(u',x)) \geq \min(d(u,x)) - \frac{w_0}{2} > 0$ for $u' \in U', x \in \mathbf{w}$. Furthermore, since the original points of $U$ are correctly classified by $\mathbf{w}$ with minimum distance $\frac{w_0}{2}$ from $\mathbf{w}$, then $U' \cap \mathcal{A}' \subset \mathbf{w}^-, U' \cap \mathcal{B}' \subset \mathbf{w}^+$, and therefore $\mathbf{w}$ strictly separates $U'$. Since this is true for all $U'$, then by Kirchberger's Theorem there exists hyperplane $\mathbf{w}'$ that strictly separates $\mathcal{A}' \cup \mathcal{B}'$. It is clear that that $\mathbf{w}'$ also strictly separates $\mathcal{A} \cup \mathcal{B}$. Additionally, we have that $\min(d(a,x)) \geq \frac{w_0}{2}$, and $\min(d(b,x)) \geq \frac{w_0}{2}$ for all $a \in \mathcal{A}, b \in \mathcal{B}, x \in \mathbf{w}'$. $\qquad\square$

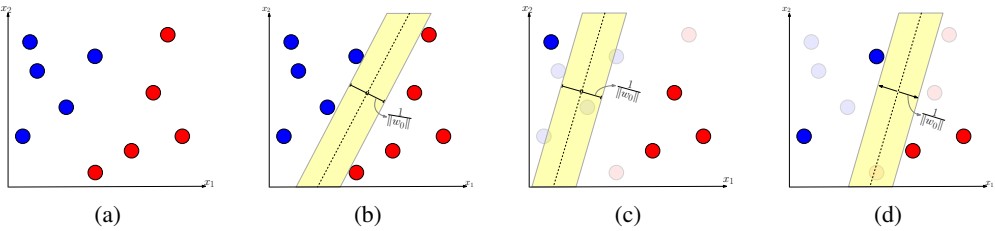

Figure 2: (a) Original set. (b) Original set with $w_{min}$. (c) and (d) $(d+2)$-tuples linearly separable with margin $w_0$.

## 3.2 FRACTIONAL KIRCHBERGER THEOREM

Here, we will prove Theorem 3 by harnessing the point-hyperplane duality relation (as in Section 2.1) in conjunction with the Fractional Helly Theorem 2.2. Our objective is to establish a vital connection: the lower bound on set intersections, as provided by the Fractional Helly Theorem, is equivalent to a lower bound for the number of linearly separable points in a binary class dataset. The essential connection is made by observing that for a point $p$ of a given class, the union of all separating hyperplanes that correctly classify $p$ forms a convex set under the duality transformation. The problem of proving the existence of a common separating hyperplane is thereby re-framed as a problem of proving the existence of a common point of intersection amongst convex sets in the dual space, for which the solution is provided by the Fractional Helly Theorem. Before proving the main result, we formally establish the correspondence between common intersections in the dual space and separating hyperplanes (as demonstrated in the example in Section 2.1). Throughout this paper, we denote the two classes of data points as $\mathcal{A}$ and $\mathcal{B}$. We make a simplifying assumption, without loss of generality, that our objective is to find a hyperplane placing $\mathcal{A}$ in its corresponding negative open half-space and and $\mathcal{B}$ in the positive open half-space.

**Lemma 1.** *For any $A \subseteq \mathcal{A}$ and $B \subseteq \mathcal{B}$, $A$ and $B$ are strictly linearly separated by a hyperplane $h$ in $E^d$ if and only if the corresponding dual point $h^*$ in the transformed space $E^{d+1}$, lies in the intersection of all negative dual transforms of points in $A$, as well as, in the intersection of positive dual transforms of points in $B$, i.e., $h^* \in \bigcap_{a \in A} \mathcal{D}^-(a), \quad h^* \in \bigcap_{b \in B} \mathcal{D}^+(b)$.*

*Proof.* If the transformed point $h^* \in \bigcap_{a \in A} \mathcal{D}^-(a)$, in $E^{d+1}$, then by Definition 4, we have that, each point $a \in A$, $a_1 h_1 + a_2 h_2 + \cdots + a_d h_d + h_{d+1} < 0$. Similarly, for each point $b \in B$, we have that, $b_1 h_1 + b_2 h_2 + \cdots + b_d h_d + h_{d+1} > 0$. Recall that $h = x_1 h_1 + x_2 h_2 + \cdots + x_d h_d + h_{d+1} = 0$, in $E^d$. Thus, $A \subset h^-$ and $B \subset h^+$, in the primal space, and therefore, $h$ strictly separates $A \cup B$. Alternatively, if $A \cup B$ are strictly separated by a hyperplane $h'$, then for $h' = \{x \in E^d : h_1 x_1 + h_2 x_2 + \cdots + h_d x_d + h_{d+1} = 0\}$, Definitions 2 and 3 assert that for all $a \in \mathcal{A}, h_1 a_1 + h_2 a_2 + .. + h_d a_d + h_{d+1} < 0$, and for all $b \in \mathcal{B}$ $h_1 b_1 + h_2 b_2 + .. + h_d b_d + h_{d+1} > 0$.

Then by Definition 6, $h'^* \in \mathcal{D}^-(a)$ and $h'^* \in \mathcal{D}^+(b)$. In words, if point $a$ is below hyperplane $h'$ and point $b$ is above hyperplane $h'$ in $E^d$ then hyperplane $a^*$ is below point $h'^*$ and hyperplane $b^*$ is above point $h'^*$ in $E^{d+1}$. Therefore, $h'^*$ is a point in the intersection of negative/positive halfspaces bounded by $a^*$ and $b^*$. $\qquad\square$

We are now prepared to prove one of our key findings, which establishes a precise lower bound on the misclassification error for any linear classifier. This result directly relates to a fundamental aspect of support vector machines.

*Proof.* **Theorem 3 (Fractional Kirchberger)**: The duality transform may be applied to $\mathcal{A} \cup \mathcal{B}$ to obtain the family of halfspaces $C = \mathcal{D}^-(\mathcal{A}) \cup \mathcal{D}^+(\mathcal{B})$. Thus, $C$ contains $n$ halfspaces of $E^{d+1}$ that are in one to one correspondence with the points of $\mathcal{A} \cup \mathcal{B}$. Let $f$ denote an arbitrary $(d+2)$-member subset of $\mathcal{A} \cup \mathcal{B}$. By Lemma 1, if $f$ admits a strict linear separation, then $\bigcap \mathcal{D}^-(f \cap \mathcal{A}) \cap \mathcal{D}^+(f \cap \mathcal{B})$ is non-empty. If there are $\alpha\binom{n}{d+2}$ such $(d+2)$-member subsets of $\mathcal{A} \cup \mathcal{B}$, then there are $\alpha\binom{n}{d+2}$ intersecting $(d+2)$-tuples of $C$. It follows from the Fractional Helly Theorem, that there are at least $\beta n$ halfspaces of $C$ that share a common intersection. Then by Lemma 1, the dual of a point $h^* \in E^{d+1}$ from this intersection produces a hyperplane $h \in E^d$ that strictly separates at least $\beta n$ members of $\mathcal{A} \cup \mathcal{B}$. $\qquad\square$

Here, $\beta$ (the optimal lower bound stated in Theorem 6) is in fact an asymptotic bound that holds for all $n$. However, we may refine this bound when we wish to assess the fractional linear separability of $\mathcal{A} \cup \mathcal{B}$ for a specific value of $n$. As established in the proof of Theorem 3, linear separators correspond to intersection points among the halfspaces of $\mathcal{A} \cup \mathcal{B}$. Therefore, we can readily apply the bound from Theorem 7 to the case of fractional linear separability. Once we determine the value of $\alpha$, which signifies the fraction of strictly linearly separable $(d+2)$-tuples in $E^d$, we can conclude that there are $\alpha n$ intersecting dual halfspaces in $E^{d+1}$. Now, considering $d' = d+1$ and $k = d'+1$, Theorem 7 can be applied, resulting in:

$$\beta = \frac{r + d' + 1}{n}, \tag{1}$$

where $r$ is determined as:

$$r = \arg\max_r \left\{ r \mid \sum_{i=0}^{d'} \binom{r}{k-i}\binom{n-r}{i} < \alpha\binom{n}{k} \right\}. \tag{2}$$

In this manner, we leverage the dataset's size to establish a precise lower bound on the fractional linear separability of $\mathcal{A} \cup \mathcal{B}$. To demonstrate the tightness of the lower bound provided by Theorem 3 in relation to the optimal linear separator, we conducted a series of experiments. In these experiments, we randomly placed points in a $d$-dimensional hypercube, assigning each point a random label of either $y_i = 1$ or $y_i = -1$ with equal probability. We repeated this process for $n = 20$ points across 5000 trials. For each trial, we performed and recorded the following computations:

1. We calculated $\alpha$ by examining each $(d+2)$-tuple of points to test for linear separability.
2. Using $\alpha$, we we derived the theoretical lower bound of $\beta$ using equations equation 1 and equation 2.
3. We determined the true value of $\beta$ by identifying an optimal linear separator minimizing the misclassification count.

The procedure was conducted separately for $d = 2$ and $d = 3$, and the aggregated results are presented in Figure 3.

### 3.3 Soft-Margin SVM Classification

In this subsection, we utilize the Fractional Kirchberger Theorem to establish a fractional counterpart to Theorem 2. Essentially, Theorem 4 provides a lower bound on the performance of a soft-margin SVM when not all samples from the dataset $\{(X_i, y_i)\}$ can be accurately linearly classified with a margin of $w_0$. As previously, we maintain the notations $\mathcal{A}$ and $\mathcal{B}$ to denote the two classes within data.

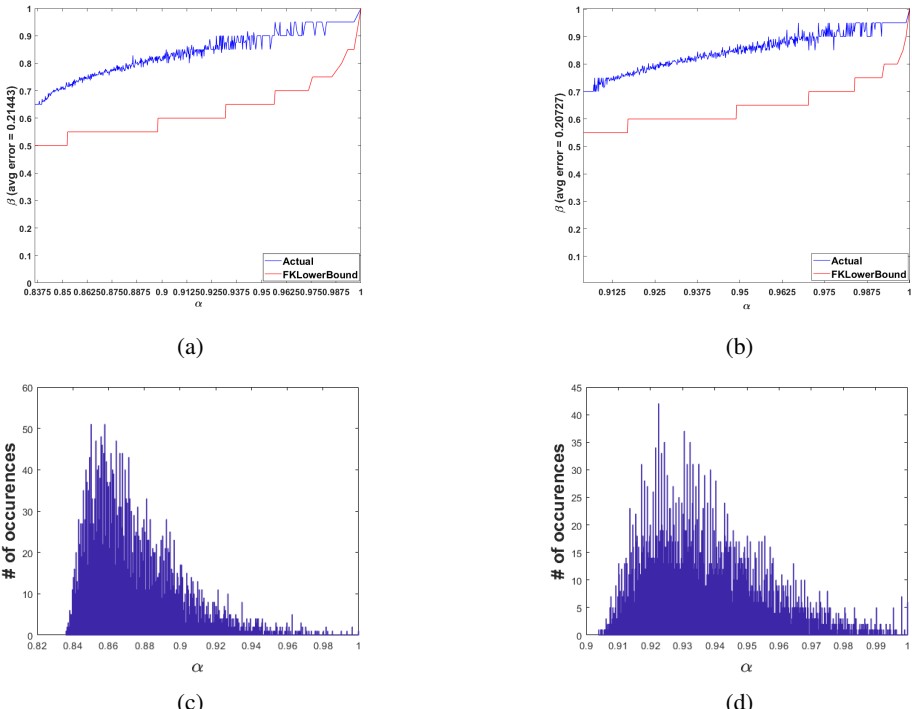

Figure 3: (a) $\beta$ comparison in $E^2$. (b) $\beta$ comparison in $E^3$. (c) $\alpha$ histogram for $E^2$. (d) $\alpha$ histogram for $E^3$.

*Proof.* **Theorem 4 (SVM Fractional Kirchberger)**: Since each of the $\alpha\binom{n}{d+2}$ of the $d+2$-member sets of $\mathcal{A} \cup \mathcal{B}$ admit a linear separator with margin $w_0$, they are linearly separable, then Theorem 2 implies the existence of $\beta$, for which there exists some $U \subset \mathcal{A} \cup \mathcal{B}$, with $|U| \geq \beta n$ such that $conv(U \cap \mathcal{A}) \cap conv(U \cap \mathcal{B}) = \emptyset$. To simplify the following argument, we may assume that each $(d+2)$-member subset of $\mathcal{A} \cup \mathcal{B}$ that are not separable with margin $w_0$ are also not linearly separable. Observe that the lower bound of $\beta$ would remain valid with or without our simplifying assumption, which is to say that the $(d+2)$-tuples of $\mathcal{A} \cup \mathcal{B}$ that are separable with margin $w_0$ but are possibly linearly separable have no representation in $U$. By this assumption, if a set of points is linearly separable, it is linearly separable with margin $w_0$. Since $U$ is linearly separable, then by Kirchberger's Theorem, every $(d+2)$-member subset of $U$ must be linearly separable and therefore by our assumption must be separable with margin $w_0$. Then we observe that there are some closest pair of points $(a_{min}, b_{min}) : \{a_{min} \in U \cap \mathcal{A}, b_{min} \in U \cap \mathcal{B}\}$ such that $d(a_{min}, b_{min}) > \delta$, and the remainder of the proof follows identically to that of Theorem 4. $\qquad\square$

We employ the Fractional Kirchberger Theorem to obtain $U$ because the $(d+2)$-member subsets of $U$ differ from the $(d+2)$-tuples of $\mathcal{A} \cup \mathcal{B}$ that are separable with margin $w_0$. Theorem 4 necessitates that linear separation with with margin $w_0$ holds for the entire family of $(d+2)$-tuples of the set under consideration.

## 4 FRACTIONAL HYPERSPHERE CLASSIFICATION

In this section, we present the paper's final result: a fractional Kirchberger-type theorem that establishes a lower bound on the performance of a hypersphere classifier when applied to a specific dataset. This lower bound is based on the classifier's performance with samples of size $d + 3$. Conditions for achieving strict spherical separability of the entire dataset are well-known in the literature Lay (2007; 1971); Simons & Trapp (1974), and we extend these results to the fractional setting. We begin by providing a formal definition of spherical separability.

**Definition 5.** *(Spherical Separability)- Given point sets $\mathcal{A}$, $\mathcal{B}$ in $E^d$. $\mathcal{A} \cup \mathcal{B}$ are strictly separable by hypersphere $h_s = \{x \in E^d : \|x - p\| = \gamma\}$ if for $a \in \mathcal{A}$, $\|a - p\| < \gamma$ and for $b \in \mathcal{B}$, $\|b - p\| > \gamma$, or vice versa.*

Our proof method involves stereographically projecting $E^d$ onto a tangent hypersphere of $E^{d+1}$ (as Figure 4 illustrates), effectively transforming the problem into one of linear separation in $E^{d+1}$ (hence the $d + 3$ requirement). We consider the point set $\mathcal{A} \cup \mathcal{B}$, where not all of its $(d + 3)$-member subsets exhibit strict spherical separability. To extend this result to the fractional case, we demonstrate that when only $\alpha\binom{n}{d+3}$ of $(d + 3)$-point samples can be correctly classified by a hypersphere, we can apply the Fractional Helly Theorem to the dual of the projected points. This yields a lower bound on the size of the largest subset with a common intersection ($\beta n$). Through the duality transformation, this lower bound on the intersection number in the dual space corresponds to a lower bound on the number of points in the original dataset that are accurately classified by a hypersphere.

*Proof.* **Theorem 5 (Fractional Hypersphere Separation):** Let $T$ be a subset of $d + 3$ points in $E^d$. Consider the embedding of $E^d$ into a hyperplane $h$ of $E^{d+1}$. Let $S$ be a $(d + 1)$-dimensional hypersphere tangent to $h$ at an arbitrary point $p$. Let $p_0$ denote the antipodal point to $p$. Then there is a bijective map, $\pi$, that sends each point $z \in E^d$ to $\{S/p_0\} \cap r$, where $r$ is the ray originating from $z$ and passing through $p_0$. If $T$ is strictly spherically separable in $E^d$, then a $d$-dimensional hypersphere $h_s$ exists such that all points in $T \cap \mathcal{A}$ lie in $h_s^{int}$ and all points in $T \cap \mathcal{B}$ lie in $h_s^{ext}$, where $h_s^{int}$ and $h_s^{ext}$ denote the interior and exterior of $h_s$, respectively. Considering the projections of $T$ and $h_s$ onto $S$, it follows that $\pi(h_s)$ is contained in the intersection of $S$ with a hyperplane $H_s \in E^{d+1}$ such that $\pi(h_s^{int})$ and $\pi(h_s^{ext})$ are strictly linearly separated by $H_s$. Observe that the projections of a set of $d + 3$ points in $E^d$ corresponds to a set of $d' + 2$ points in $E^{d'}$ for $d' = d + 1$. Since there are $\alpha\binom{n}{d'+2}$ such subsets of projected points that are strictly linearly separable in $E^{d'}$, then by Theorem 3 there exists a hyperplane $H'_s \subset E^{d'}$ that strictly linearly separates the projections of $\beta n$ members of $\mathcal{A} \cup \mathcal{B}$. Thus, $\pi^{-1}(H'_s)$ is a hypersphere in $E^d$ that satisfies the claim. $\qquad\square$

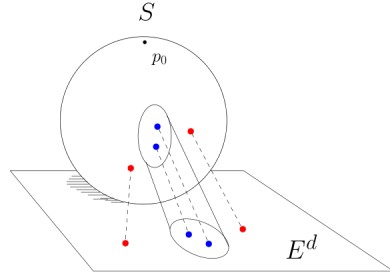

Figure 4: Stereographic projection of $E^d$ onto $S$.

## 5 CONCLUSION

Through the lens of duality, we have unveiled the remarkable potential of the Fractional Helly Theorem to establish lower bounds on fractional linear separability, fractional separability with margin, and fractional separation by hypersphere. Notably, the bounds deduced from Theorem 2 and Theorem 4 find direct, real-world applications in hard-margin and soft-margin SVMs, respectively. One paramount aspect of our work is the deterministic nature of these lower bounds. Each bound is obtained as a direct combinatorial implication of the given classifier's performance on small subsets of the dataset $\{(X_i, y_i)\}$, offering a practical and qualitatively distinct alternative to VC-dimension-based performance analysis. Moreover, this approach takes into account dataset distribution while maintaining manageable local computation requirements with fixed sample cardinality. The fusion of combinatorial methods with machine learning approaches will afford a fresh perspective on evaluating classification potential. The practical implementation of these bounds and their integration into real-world machine learning systems opens exciting avenues for further research and innovation.

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
