# OpenReview forum: "Quantifying Classification Performance through Combinatorial Geometry and Localized Data Analysis"
_ICLR.cc/2024/Conference — ICLR 2024 Conference Withdrawn Submission_

### Official Review · Reviewer_RpW5 · 2023-10-25

**Soundness:** 3 good
**Presentation:** 2 fair
**Contribution:** 2 fair
**Rating:** 5
**Confidence:** 3

**Summary:**

The authors provide a characterization of linear separability of a dataset by its local separability. Based on Kirchberger Theorem, the authors first provide an extension of Kirchberger Theorem to SVMs. Utilizing the Fractional Helly Theorem, the authors also generalized the results to fractional case. Finally a result for spherical separability is also provided.

**Strengths:**

This paper generalize Kirchberger Theorem to SVM and fractional setting, which is of good novelty.

**Weaknesses:**

1. The results seems not very related to machine learning. It is unclear to me under what scenarios in machine learning we can apply the results.

2. The major usage of VC dimension in machine learning context is to derive generalization bounds, which seems not to be the main purpose of this paper (measure the separability of a dataset). The authors emphasize VC dimension in abstract and introduction, which makes me a little bit confusing.

**Questions:**

Other than SVMs and spherical classifiers, is it possible to show the same type of results for other non-linear classifiers?

---

### Official Review · Reviewer_VSAx · 2023-10-29

**Soundness:** 4 excellent
**Presentation:** 4 excellent
**Contribution:** 2 fair
**Rating:** 5
**Confidence:** 4

**Summary:**

The paper studies the problem of connecting global guarantees and local guarantees for linear separability problem.
Formally, we have the following problem.
Suppose we are given $n$ points in $\mathbb{R}^d$ where each point associates with a $\pm 1$ label and we would like to check if they are linearly separable, i.e. there exists a hyperplane such that all points with $+1$ are in one side of the hyperplane and all points with $-1$ are in the other side.
However, when $n$ is large, checking linear separability for the entire dataset may not be feasible.
Now, if we manage to check whether all subset of size $d+2$ are linearly separable, then we can conclude whether the entire dataset is also linearly separable.
The authors extended to the setting that the point set is linearly separable with a margin and a fraction of points is linearly separable (with a margin) and gave an affirmative answer.

**Strengths:**

- The paper is well-written.
The readers from all levels of expertise should be able to understand this paper.

- The paper provides an interesting connection between machine learning and geometry.
I believe this could be a good starting point for this line of research.

**Weaknesses:**

- In terms of the techniques, it seems that the crux of the proofs is Kirchberger theorem which is a well-known result and the proofs then follow rather straightforward reduction to this theorem.
I am not sure if there are any fundamentally new techniques introduced.

- If I am not mistaken, we still need to check the linear separability with margin of (a fraction of) all possible subsets of size $\sim d$ to conclude anything for the entire dataset and it has an exponential dependence on $d$.
When the dataset is in high-dimensional spaces, I am not sure how the authors' result fundamentally helps.

**Questions:**

na

---

### Official Review · Reviewer_re7m · 2023-10-31

**Soundness:** 4 excellent
**Presentation:** 3 good
**Contribution:** 1 poor
**Rating:** 3
**Confidence:** 4

**Summary:**

The paper introduces a new way to quantify classification performance of SVM models on a particular sample by a fraction number $\beta$ of sample than are linearly separable (where $\beta=1$ for complete separation). To this end, the authors propose to find a lower bound for $\beta$ via Kirchberger Theorem (1903), which characterizes the global point separation by local point separations. Specificly, the authors prove that $\beta \geq 1-(1-\alpha)^{1/(d+1)}$ where $d$ is the dimension and $\alpha$ is the fraction of all $(d+2)$ subsets that are linearly separable.

**Strengths:**

The authors propose an interesting idea of using combinatorial geometry to study the classification performance of SVM. In contrast to VC-type bound, the performance depends on the data configuration, which can be useful in many cases. Moreover, the presentation on the combinatorial geometry is clear and easy to understand.

**Weaknesses:**

- From the introduction, the authors seem to imply that the fractional separability ($\beta$) is proposed as an alternative to VC, however, I don't see how they are comparable to each other, as VC can be used to derive generalization bound that is independent of data distribution. The fractional separation, however, only looks at the geometry of the sample and ignores the population's distribution. This means that the fractional separability can only be used to quantify the classification power on the current data points and nothing beyond that.
- It is computationally expensive to obtain the proposed lower bound of $\beta$, since we have to check linear separability on $\binom{n}{d+2} = \Theta(n^{d+2})$ subsets. Even with parallelized computation I think the amount of computation is still too much.
- Conversely, if one wants to certify that $\beta \geq 0.8$ with e.g. $d=10$, they have to check linear separability for at least $1-0.2^{11}$ fraction of all subsets, which is close to all possible subsets of size $d+2$.
- It should be easier to check the linear separability by just run the convex program for soft SVM on the whole dataset and check the slack variables.
- On the flip side, the proposed method can be used to check if the data is *not* linearly separable---we just need to find a single subset of size $d+2$ that cannot be linearly separable. This approach has been discussed in [1], and I think it would be fruitful to come up with a sampling scheme that leads to a non-linearly separable subset earlier in the search.

To summarize, even though the paper's contribution is currently inadequate, the results in this paper could lead to something useful. In particular, we should be able to derive from the lower bound some kind of generalization bound (e.g. the *population* is linearly separable with high probability) or some finite-subset type bound (e.g. assuming that the data lies in a convex set, we can check that the sample is linearly separable with high probability by subsampling uniformly for $k \ll \binom{n}{d+2}$ times).

Reference
[1] Haghighatkhah, P., Meulemans, W. ., Speckmann, B., Urhausen, J., & Verbeek, K. (2022). Obstructing Classification via Projection. Computing in Geometry and Topology, 1(1), 2:1–2:21. https://doi.org/10.57717/cgt.v1i1.8

Minor typos:
- Paragraph next to Theorem 4: Kirhberger -> Kirchberger
- Page 3: the the -> the
- Page 4: the this -> the (or this)

**Questions:**

See Weaknesses.

---

### Official Review · Reviewer_t2D4 · 2023-11-08

**Soundness:** 2 fair
**Presentation:** 2 fair
**Contribution:** 3 good
**Rating:** 6
**Confidence:** 3

**Summary:**

This work harnesses the underlying combinatorial geometry of data to address the problem of establishing realistic bounds on a model’s classification power. It establishes optimal bounds on the training error of a linear classifier, proves an optimal bound on the margin of Support Vector Machines (SVMs) in terms of performance of SVMs on (d + 2) sized subsets of data, and extends these results to a non-linear classifier employing hypersphere boundary separation.

**Strengths:**

This paper makes some contributions in classification learning: It proves realistic bounds on a model’s classification power. This work introduces conditions that rely on local computations performed on small data subsets to determine the global performance of Classifiers and establishes optimal bounds on the training error of a linear classifier.  It also proves an optimal bound on the margin of SVMs in terms of performance.

**Weaknesses:**

1.The logic of the Introduction Section can be improved to make it easier to read.

2.The words on the figures are too small to read clearly.

3.There are some writing errors. For example: on page 7, “we we derived the theoretical lower bound of β using equations equation 1 and equation 2.”

4.If the research of this work can be further improved, it may be more suitable for this conference.

**Questions:**

See "Weaknesses"

---

### Official Review · Reviewer_VFVG · 2023-11-08

**Soundness:** 4 excellent
**Presentation:** 3 good
**Contribution:** 3 good
**Rating:** 5
**Confidence:** 5

**Summary:**

This paper presents a series of theorems on convex classifiers that improves the theoretical understanding of the classifiers in several ways: first, the results in this paper put the local to global generalization of classifiers performance on a given dataset, on a rigorous mathematical basis, second, it extends these results to several common families of classifiers, and it provides an optimal bound.

**Strengths:**

The strengths of this paper were its originality in the use of geometric methods, the quality of the results that comes from proof based methods, and its significance, in that it is exploring fundamental problems of the classifier, rather than observing properties of the classifier on one or several datasets.

The use of hyperplane duality to prove formal results is a novel technique that provides real insights into the performance of classifiers, independent of the data used. The methods used were proof based, which is itself fairly novel in the machine learning realm and makes the results higher quality than if they had just run against some scavenged datasets.

**Weaknesses:**

There were several minor weaknesses in this paper:

First was the significance of the supporting data experiments, which were weak, relying only on synthetic data and given very little attention in the general presentation. In general, the paper would be stronger with some discussion of what data achieves the discovered bounds, and is that data present in real world data. It wouldn't have to be a giant dataset, even data as simple as IRIS or the housing price datasets would illustrate some interesting complexity beyond that of synthetic data

Second, which is mostly superficial, was arguing for importance of the results. The results are great theoretically, but some discussion why linear and convex classifiers are relevant in a world full of deep models would make the paper stronger and the results more interesting. It needn't be a super detailed argument, even something as simple noting that linear classifiers are the most common first model used in practice, and there are ML practitioners who go through life trying linear classifiers on every problem and if a linear classifier doesn't work, they move on to a new problem.

Third was the presentation. The paper was lacking a detailed description of SVMs and Hyphersphere classifiers, which detracted from the clarity of the presentation. Additionally, I found the proofs in section 3 were difficult to follow, both in terms of the text-heavy proof style, their location overall in the document, and the basic, slightly repetitive structure of the argument. My feeling is that the authors had only one theorem level result in this document (theorem 6) and that everything else followed as a corollary. Moreover, I think that stating that theorem in convex, fractional terms is the most useful level of generality, and makes it more clear that the results are all basic variations on a theme.

**Questions:**

Suggestions are largely included in weaknesses.

The big question I had on on this paper that didn't come up in the weaknesses, is I'm not sure why this paper is belongs in a machine learning paper. It feels like it this would be at home in an applied math journal as well. Part of this is that the related work does not draw significantly from the machine learning lit. And, it seems like much of related work is in math journals, including some of the related work that isn't cited, like this one:
- Needell, Deanna. “Large Data Analysis and Lyme Disease.” Notices of the American Mathematical Society 66, no. 6 (January 2019): 8–15.

One, somewhat minor question was on, the hypersphere classifiers. It felt like these were included because it was an easy generalization of the results, but this also seems like an extension that is important to ball-tree types of data structures and knn style classifiers. My question is a) do these results apply in those areas? If so, I think you would have a more compelling for the significance of these results.

A second potential extension that I am curious about is, it seems like these results can all be phrase in pure oriented matroid terms, and using duality in that form might also let you apply these results to graph based classifiers.